# An Acid-Sensitive Bone Targeting Delivery System Carrying Acacetin Prevents Osteoporosis in Ovariectomized Mice

**DOI:** 10.3390/ph16010002

**Published:** 2022-12-20

**Authors:** Xiaochen Sun, Chenyu Song, Chenxi Zhang, Chunlei Xing, Juan Lv, Huihui Bian, Nanning Lv, Dagui Chen, Xin Dong, Mingming Liu, Li Su

**Affiliations:** 1School of Medicine, Shanghai University, Shanghai 200444, China; 2Institute of Translational Medicine, Shanghai University, Shanghai 200444, China; 3Department of Orthopedics, Lianyungang Second People’s Hospital, Lianyungang 222006, China; 4Lianyungang Clinical School, Xuzhou Medical University, Lianyungang 222006, China

**Keywords:** osteoporosis, autophagy, acid-sensitive linker, bone-targeting peptide, postmenopausal osteoporosis

## Abstract

One effective treatment for postmenopausal osteoporosis is to inhibit osteoclasts and subsequent bone resorption. In our study, we demonstrated that acacetin, a flavone with potential therapeutic effects in infections, cancers, and several metabolic disorders, inhibited osteoclast differentiation and bone resorption in vitro. For improving the efficacy of acacetin in vivo, we developed an acid-sensitive bone-targeting delivery system composed of an acid-sensitive linker (N-ε-maleimidocaproic acid hydrazide, EMCH) for ensuring an effective release of acacetin at the site of action and a hydrophilic aspartic acid hexapeptide ((Asp)6, D6) as the effective bone targeting agent. Our results revealed that Acacetin-EMCH-D6 specifically bound to the bone surface once administrated in vivo, prolonged the retention time in bone and released acacetin at the osteoclastic bone resorption sites where the acidity is higher. We further demonstrated that, in ovariectomy-induced osteoporosis mice, treatment with Acacetin-EMCH-D6 inhibited osteoclast formation and increased trabecular bone mass. On the contrary, neither acacetin nor EMCH-D6 with the same dosage alone showed significant anti-osteoporosis effects in vivo. Mechanistically, targeted delivery of acacetin to the bone resorption sites by Acacetin-EMCH-D6 inhibited autophagy through activating PI3K/AKT/mTOR pathway in osteoclasts, while the activation of autophagy by rapamycin partially reversed the inhibitory effects of acacetin in vitro and in vivo. In summary, our study, for the first time, showed that the acid-sensitive bone-targeting delivery system carrying acacetin was effective for the treatment of postmenopausal osteoporosis. Thus, targeted delivery of acacetin using Acacetin-EMCH-D6 to bone resorption sites is a promising therapy for osteoporosis.

## 1. Introduction

Osteoporosis following the estrogen withdrawal of menopause is a serious global clinical problem with a significant increase in the risk of fractures [1]. Supplementation of estrogen is not recommended for the long-term treatment of postmenopausal bone loss because of the increased risk of breast cancers and cardiovascular events [2,3]. Pure compounds isolated from natural products are emerging as potential therapeutics for postmenopausal osteoporosis [4]. Acacetin is a di-hydroxy and mono-methoxy flavone widely distributed in various plants including the Acacia tree, black locust, and Chrysanthemum [5]. Studies revealed that acacetin exhibited immense therapeutic potential in infections, obesity, cancer, and cardiovascular disorders [6,7]. It was recently reported that acacetin mitigated RANKL-induced osteoclastic differentiation from macrophages in vitro [8,9], suggesting its potential benefit against osteoporosis. However, it was reported that the concentration of acacetin in plasma rapidly declined after intravenous administration and was removed following a terminal half-life of 1.48 h [10]. Because of insufficient blood supply in bone with aging, bioactive drugs would hardly be delivered to bone lesions, which led to limited clinical utilization and therapeutic efficacy. Lin et al. provided evidence that treatment with 20 mg/kg/day acacetin for 8 weeks protected against bone loss in ovariectomized (OVX) mice [11]. However, we did not observe its anti-osteoporotic effects at the dose of 5 mg/kg/3 day in OVX mice (Appendix A). For improving the efficacy of acacetin treatment, we designed an acid-sensitive bone targeting delivery system (Figure 1A), and the design features were the use of the hydrophilic aspartic acid hexapeptide as the effective bone targeting agent and an acid-sensitive bond for ensuring an effective release of acacetin at the site of action.

In this work, we tested the therapeutic efficacy of the novel acid-sensitive bone-targeting delivery system carrying acacetin against osteopenia in OVX mice. In addition, we investigated the underlying molecular mechanism of the inhibitory effects of acacetin on osteoclastic differentiation and bone resorption.

## 2. Results

### 2.1. Design and Chemical Synthesis of Acacetin-EMCH-D6

In order to design bone-targeting delivery systems properly, acid-sensitive linkers should be considered. According to literature reports, osteoclasts degrade HA in an acidic microenvironment to achieve bone remodeling processes. At present, acid-sensitive structures used for delivery system construction mainly include tert-butyloxycarbonyl, paramethoxybenzyl, orthoester, acetal and ketal, etc. Among them, the hydrazone bond is a chemical bond formed by the coupling of carbonyl compound and hydrazine compound, which can undergo a reverse reaction to release prototype substrate under acidic conditions. Considering the unique unsaturated carbonyl group in the structure of acacetin, we selected (6-maleimide hexyl) hydrazone derivatives as an acid-sensitive smart linker in this work.

The next step is the identification of bone-targeting peptides. Bone tissue, as a special connective tissue, consists of a variety of cells and intercellular bone matrix, with high inorganic salt content which is mainly composed of HA. In recent years, more and more peptide sequences with excellent bone-targeting ability have been discovered. Compared with bone-targeting small molecules, these peptides have more specific mechanisms, high affinity and low side effects. Considering that acacetin has an excellent inhibitory effect on osteoclasts, we selected Asp-Asp-Asp-Asp-Asp-Asp (Asp6, D6), a medium-length polyaspartic acid with HA affinity comparable to tetracycline and calcein, as the targeted peptide in this work.

In general, we coupled acacetin with polyaspartic acid-based bone-targeting peptide through derivatives of 6-maleimide hexyl hydrazones to the bone targeting acacetin-EMCH-D6 (Figure 1A). As shown in Figure 1B, acacetin (1) and commercially available N-ε-maleimidocaproic acid hydrazide (EMCH, 2) were coupled to obtain intermediate (3, Acacetin-EMCH, Appendix A) via catalysis sodium p-toluene sulfonate with a yield of 74%. On the other hand, Wang resin used as solid phase support on resin peptide (4) was provided using 6-chloro-1H-benzotriazol-1-yloxy (dimethylamino)-N,N-dimethylmethani-minium hexafluorophosphate (HCTU) and N,N-diisopropylethylamine as condensation reagents by sequentially coupling FMOC-protected aspartate and cysteine with standard solid-phase peptide synthesis (SPPS). The resin peptide (4) was subjected to an acid cleavage cocktail to afford the crude target peptide (5). After purification by preparative reverse phase chromatography and subsequent lyophilization, the pure product was obtained as a lyophilized powder with a chemical yield of 51% (Appendix A). The intermediate (3) and (5) then undergo a thiolene click reaction to obtain the target Acacetin-EMCH-D6 with a high yield (Appendix A). Besides, as shown in Figure 1C, EMCH (2) and intermediate peptide (5) also undergo a thiolene click reaction to obtain the EMCH-D6 (Appendix A).

### 2.2. Drug Release Properties and Hydroxyapatite Affinity of Acacetin-EMCH-D6

We incubated Acacetin-EMCH-D6 at pH 5.5 and analyzed the release of acacetin by HPLC (Figure 1D, Appendix A). At pH 5.5, all of the bioactive compound was released after 8 h of incubation. However, only 6.43% of acacetin was released after 8 h of incubation at pH 7.4. When acacetin was incubated with hydroxyapatite (HA), only very weak binding of acacetin to HA could be observed. In contrast to acacetin, the Acacetin-EMCH-D6 was rapidly bound to HA (Figure 1E, Appendix A).

### 2.3. Acacetin Inhibited Osteoclast Differentiation and Bone Resorption

Incubation with EMCH-D6 (0, 6.25, 12.5, 25, 50, 100 μM) for 48 h (Figure 2A) and 96 h (Figure 2B) did not affect the cell viability in cultured BMMs. Incubation of BMMs with acacetin did not affect cell viability at the dose of 6.25, 12.5, and 25 μM, but reduced cell viability when the dose was more than 50 μM.

In cultured BMMs stimulated with RANKL and M-CSF, treatment with acacetin significantly diminished the number of TRAP-positive osteoclasts (nuclei ≥ 3, Figure 2C,D) and area of bone resorption pits (Figure 2E,F) in a dose-dependent manner.

In addition, treatment with acacetin suppressed mRNA expression of TRAP (Figure 2G), NFATc1 (Figure 2H), and CTSK (Figure 2I) in BMMs stimulated with RANKL and M-CSF.

Treatment with EMCH-D6 did not affect osteoclast differentiation and bone resorption in BMMs when stimulated with RANKL and M-CSF.

### 2.4. Acacetin-EMCH-D6 Treatment Inhibited OVX-Induced Bone Loss in Mice

A decline in estrogen levels in OVX mice leads to increased bone resorption as well as reduced bone mass. For the in vivo distribution assay, mice were intraperitoneally injected with FITC-labeled Acacetin-EMCH (17.5 μmol/kg) and FITC-labeled Acacetin-EMCH-D6 (17.5 μmol/kg). In the liver, lung and kidney, fluorescence intensity was detected at 30 min, but little residual fluorescence was detected 2 h after injection (Figure 3A). In the lower limb and spine of the Acacetin-EMCH-D6-injected group, fluorescence intensity was maintained at a high level 2 h after injection; 2 h after injection, little residual fluorescence was detected in the lower limb and spine of Acacetin-EMCH injected group. Our results indicated that EMCH-D6 facilitated acacetin targeting bone and prolonged the retention time in bone.

As shown in H&E staining results (Figure 3B), the OVX group exhibited considerable bone trabecular loss in the proximal tibia. Treatment with Acacetin-EMCH-D6 raised the trabecular density and thickness. TRAP staining results revealed that the number of TRAP-positive cells increased dramatically in the OVX group (Figure 3C,D), which was significantly reversed by treatment with Acacetin-EMCH-D6. Treatment with Acacetin or EMCH-D6 alone did not affect the histological changes in the proximal tibia after ovariectomy.

As compared to the control mice, ovariectomy led to increased CTX-1 levels (Figure 3E) and reduced estrogen levels (Figure 3F) in plasma. Treatment with Acacetin-EMCH-D6 reduced plasma CTX-1 levels in OVX mice. Treatment with acacetin or EMCH-D6 had no significant effect on plasma CTX-1 levels in OVX mice. Interestingly, both Acacetin-EMCH-D6 and acacetin significantly raised plasma estrogen levels in OVX mice.

To determine the effect of Acacetin-EMCH-D6 in OVX-induced bone loss, proximal tibia and L5 vertebrae were subjected to micro-CT analysis. Micro-CT with the 3D reconstruction of the proximal tibia and L5 vertebra revealed significant bone loss within vehicle-treated OVX mice (Figure 4A,F). In Acacetin-EMCH-D6-treated mice, more trabecular bone was observed. Furthermore, the Result of BMD (Figure 4B,G), BV/TV (Figure 4C,H), Tb.N (Figure 4D,I), and Tb.Sp (Figure 4E,J) showed significant increases in bone resorptive function in the vehicle-treated OVX group, which was partly reversed by treatment with Acacetin-EMCH-D6. Treatment with acacetin or EMCH-D6 had no significant effect on BMD, BV/TV, Tb.N, or Tb.Sp.

### 2.5. Acacetin Inhibited Autophagy in Osteoclast and Rapamycin Partly Abolished the Beneficial Effect of Acacetin on Osteoclast Differentiation and Bone Resorption

Stimulation of BMMs with RANKL and M-CSF reduced the phosphorylation of PI3K, AKT and mTOR and enhanced the protein levels of LC3II (relate to autophagy), which was reversed by treatment with acacetin (Figure 5). Our results indicated that acacetin activated PI3K/AKT/mTOR pathway and inhibited osteoclast autophagy.

Co-treatment with rapamycin, an activator of autophagy, abolished the inhibitory effect of acacetin on the protein levels of LC3II (Figure 6A).

NFATc1 and c-Fos are key transcription factors that regulate osteoclast differentiation [12,13]. In BMMs stimulated with RANKL and M-CSF, acacetin downregulated protein expression of NFATc1 and c-Fos, which were partially reversed by co-treatment with rapamycin (Figure 6B). Furthermore, TRAP and bone resorption assays revealed that co-treatment with rapamycin partly reversed the inhibitory effects of acacetin on osteoclast differentiation (Figure 6C,D) and bone resorption (Figure 6E,F), respectively. 

In comparison with vehicle-treated mice, microstructure parameter analysis in the proximal tibia (Figure 7A) and L5 vertebrae (Figure 7B) showed that Acacetin-EMCH-D6-treated OVX mice displayed augmentation of BMD, BV/TV, and Tb.N and reduced Tb.Sp, which was reversed by co-treatment with rapamycin, at least in part. In comparison with vehicle-treated OVX mice, Acacetin-EMCH-D6-treated OVX mice displayed a reduced number of TRAP-positive osteoclast (mature osteoclasts), which was partly reversed by co-treatment with rapamycin (Figure 7C). In comparison with vehicle-treated mice, LC3 staining results revealed that Acacetin-EMCH-D6-treated OVX mice displayed a reduced number of LC3 dots within the proximal tibia, which was partly reversed by co-treatment with rapamycin (Figure 7D).

## 3. Discussion

In this study, an acid-sensitive and bone-targeting drug delivery system was designed and used to treat osteoporosis. By treatment with Acacetin-EMCH-D6 (17.5 μmol/kg) every 3 days for 5 weeks, bone loss was significantly mitigated and trabecular bone morphometry was improved in the OVX mice model. Targeted delivery of acacetin to the bone resorption sites by Acacetin-EMCH-D6 inhibited autophagy through activating the PI3K/AKT/mTOR pathway in osteoclasts which contributed to the beneficial effects on the OVX mice model.

The acid-sensitive linker was used in our work to design bone-targeted peptide-drug conjugation (PDC) because osteoclasts degraded HA in an acidic microenvironment to achieve bone remodeling processes [14]. Acid-sensitive structures used for PDC construction mainly included: tert-butyloxycarbonyl, paramethoxybenzyl, orthoester, acetal, ketal, etc. [15]. Among them, the hydrazone bond was a chemical bond formed by coupling the carbonyl compound and hydrazine compound, which could undergo a reverse reaction to release a prototype substrate under acidic conditions. It is a typical acid-sensitive linker commonly used in biological macromolecule labeling and PDC construction [14,16,17,18,19,20,21]. Considering the unique unsaturated carbonyl group in the structure of acacetin, we selected (6-maleimide hexyl) hydrazone derivatives as an acid-sensitive smart linker in the current work. 

Bone tissue, as a special connective tissue, consists of a variety of cells and intercellular bone matrix with high inorganic salt content, which is mainly composed of HA. In previous reports, tetracycline [22], bisphosphonates [23], hydroxymalonic acid [24] and small heterocyclic compounds [25] have been frequently used to design bone-targeting drug carriers. In recent years, several peptide sequences with excellent bone-targeting ability have been developed, such as polyaspartic acid [26], Ser-Asp-Ser-Ser-Asp [27], Ser-Thr-Phe-Thr-Lys-Ser-Pro [28], (Asp-Ser-Ser)6 [29], etc. Compared with bone-targeting small molecules, these peptides had a higher affinity and lower side effects [30]. Considering that acacetin had an excellent inhibitory effect on osteoclasts, we selected (Asp)6, a medium-length polyaspartic acid with HA affinity comparable to tetracycline and calcein, as the targeted peptide in the current work. Since Acacetin-EMCH-D6 degrades into monomers acacetin and EMCH-D6 in an acidic environment, we examined the effect of acacetin or EMCH-D6 on osteoclasts in vitro.

We combined acacetin with a polyaspartic acid-based bone-targeting peptide through (6-maleimide hexyl) a hydrazone derivative linker to design the first acacetin-based acid-sensitive bone-targeting PDC. Estrogen protects against bone loss in both women and men, mainly by suppressing bone resorption. The deficiency of estrogen induces osteoporosis by enhancing the bone resorption of osteoclasts. Ovariectomy is a model of menopause which improved the knowledge of osteoporosis, cardiometabolic disorders, and inflammatory diseases. The dose and administration frequency were much lower than the previous report [11], demonstrating the application value of the acid-sensitive and bone-targeting drug delivery system in the treatment of postmenopausal osteoporosis.

Mechanistically, we found that acacetin inhibited osteoclast autophagy. Accumulating evidence revealed that autophagy dysregulation was related to the onset and progression of bone loss [31]. Autophagy promoted osteoclast podosome disassembly and enhanced their migration ability [32]. The efficient early activation of NF-κB and MAPK is necessary for the downstream induction and upregulation of transcription factors c-Fos and NFATc1. It has been regarded as the master transcription factor for osteoclast formation and function, transcriptionally regulating the expression of various osteoclast genes including *TRAP*, *CTSK*, and *NFATc1* itself. Using biochemical Western blot assays, we showed that acacetin treatment reduced the efficient induction of c-Fos and NFATc1 via the autophagy pathway. However, autophagy agonist rapamycin antagonizes the inhibitory effect of acacetin. By interacting with key components of mTORC1 or mTORC2 such as Raptor, Rictor, LST8, and SIN1, mTOR catalyzes the phosphorylation of multiple targets, such as S6K1, 4E-BP1, Akt, PKC, IGF-IR, and ULK1, regulating protein synthesis, nutrient metabolism, growth factor signaling, cell growth, migration, and autophagy, mTORC1 inactivates the formed autophagy regulatory complex (formed by ULK1 and its interaction proteins Atg13, FIP200, Atg101, etc.) by phosphorylation, affecting the biogenesis of autophagy. Our Western blot assay showed that RANKL stimulation reduced the expression of PI3K, AKT, and mTOR. However, acacetin treatment increased the expression of PI3K, AKT, and mTOR via inhibiting autophagy pathways such as the expression of LC3.

Atg5, Atg7, Atg4b and LC3 are several important proteins associated with autophagy and are closely related to forming fold margins, secretion and bone resorption function. Deletion of Atg7 in osteoclast attenuated hyperactivation of osteoclast and osteoporosis induced by estrogen withdrawal or glucocorticoid exposure in mice [33]. More importantly, we found that activating autophagy by rapamycin abolished the inhibitory effect of acacetin on osteoclast differentiation and the anti-osteoporotic effects of Acacetin-EMCH-D6 in ovariectomized mice. Therefore, the anti-osteoporotic effects of acacetin might be dependent on autophagy, at least in part. The PI3K/AKT/mTOR pathway was a critical regulator of autophagy [34]. In our work, acacetin restored the activity of the PI3K/AKT/mTOR pathway in osteoclasts, which might explain the inhibitory effect of acacetin on osteoclast autophagy.

## 4. Materials and Methods

### 4.1. Materials

Unless otherwise noted, reagents, drugs, and antibodies used in the current work were purchased from Sigma-Aldrich (St. Louis, MO, USA).

### 4.2. Preparation of Acacetin-EMCH-D6

The detailed synthesis process is described in supporting information.

### 4.3. pH-Dependent Drug Release Study

Acacetin-EMCH-D6 was dissolved in a buffer containing 4 mM sodium phosphate and 150 mM sodium chloride (pH = 7.4 or 5.5), to a final concentration of 800 μM. The samples were incubated at room temperature and measured by HPLC at times of 1, 2, 4, and 8 s.

### 4.4. Hydroxyapatite (HA) Binding Assay

Acacetin-EMCH-D6 and acacetin were dissolved, respectively, in a buffer containing 4 mM sodium phosphate and 150 mM sodium chloride (pH = 7.4), to a final concentration of 800 μM; 15 mg of HA was added and incubated in a 37 °C water bath; 20 μL aliquots were taken after 0, 10, 20, 40, and 80 min and quantified by HPLC.

### 4.5. Bone Marrow-Derived Macrophages (BMMs)

In brief, the hind limbs of C57BL/6J mice were used to isolate bone marrow cells and cultured in α-MEM containing 10% FBS and 1% penicillin-streptomycin for one day to generate BMMs as described previously [35]; 25 ng/mL M-CSF (416-ML-010/CF, R&D Systems, Minneapolis, MN, USA) and 50 ng/mL RANKL (462-TR-010/CF, R&D Systems, Minneapolis, MN, USA) were used for the preparation of osteoclasts.

### 4.6. Cytotoxicity Assay

BMMs were seeded in 96-well plates at a density of 8 × 10^3^/well and treated with acacetin and EMCH-D6 (6.25, 12.5, 25, 50, and 100 μM). The medium was changed every 2 days. After 48 h and 96 h of culture, CCK8 reagent (10 μL/well, Dojindo, Kumamoto, Japan) was added to each well. Then, the cells were incubated in darkness at 37 °C for 2 h. The absorbance was recorded using a microplate reader (Thermo Fisher Scientific, Waltham, MA, USA) at 450 nm.

### 4.7. Tartrate-Resistant Acid Phosphatase (TRAP) Staining

**Experiment** **1.**
*BMMs were cultured for 7 days with RANKL (50 ng/mL) and M-CSF (25 ng/mL) in the presence of acacetin or EMCH-D6 (0, 12.5, 25 μM). TRAP-positive multinucleated osteoclasts (≥3 nuclei) were counted using a TRAP kit (Zhejiang Zhuoteng Biological, Shaoxing, China) according to the manufacturer’s instructions.*


**Experiment** **2.**
*BMMs were induced by RANKL (50 ng/mL) and M-CSF (25 ng/mL) in the presence of acacetin (25 μM) and rapamycin (50 nmol/L) [36] for 7 days. TRAP-positive multinucleated osteoclasts (≥3 nuclei) were counted.*


### 4.8. Bone Resorption Assay

Assays for bone resorption were performed using Corning Osteo Assay Surface plate (Sigma-Aldrich, Hongkong, China). We analyzed bone resorption pits with a light microscope and quantified pits area using Image J software (NIH, Bethesda, MD, USA).

### 4.9. Real-Time Quantitative PCR (RT-PCR)

The total RNA was extracted from cells or tissues using TRIzol reagent (Thermo Fisher Scientific, Waltham, MA, USA) according to the manufacturer’s instructions. PrimeScript^TM^ RT Master Mix (RR036Q, Takara, San Jose, CA, USA) was used to reverse transcribe 1 μg of total RNA extracted into complementary DNA (cDNA) for subsequent RT-PCR reactions using qTOWER real-time PCR thermal cycler (Analytik Jena, Jena, Germany). All primer pairs of target genes were presented in Table 1.

### 4.10. Immunoblotting Assays

RIPA lysate (50 × protease inhibitor, 50 × phosphatase inhibitor) was used to extract total protein from the cells. Then, 20 μg protein or protein marker per lane (PageRuler ™ (ThermoFisher Scientific, Waltham, MA, USA) Prestained Protein Ladder 10kD-180KD or HiMark™ (ThermoFisher Scientific, Waltham, MA, USA) Pre-stained Protein Standard 31 kd-460 kd) was loaded onto an SDS gel (30% Acrylamide, 1.5 M Tris-HCl, 10% SDS, 10% Ammonium persulfate and TEMED) and subjected to electrophoresis and transferred to a polyvinylidene fluoride membrane. The membrane was blocked with 5% BSA for 4 h and incubated with the specific primary antibody overnight at 4 °C. Then, the membrane was incubated with a secondary antibody (IRDye) for 1 h at room temperature. Bio-Rad ChemiDoc MP imaging system (Bio-Rad Laboratories, Hercules, CA, USA) was used to visualize protein bands and ImageJ software was used to calculate relative protein expression based on grayscale blots.

Antibodies including p-AKT (Ser473; #4060), AKT (#4691), LC3A/B Antibody (#4108), p-mTOR (Ser2448; #5536), mTOR (#2983), c-Fos (#2250), NFATc1 (#8032), p-PI3K (#4228), PI3K (#4249), and GAPDH (#5174) were purchased from Cell Signalling Technology (Denver, MA, USA).

### 4.11. Animals

Female 6-month-old C57BL/6J mice were purchased from Cavens Laboratory Animal Co., Ltd. (Changzhou, China). According to the Animals (Scientific Procedures) Act, 1986 of the UK Parliament, Directive 2010/63/EU of the European Parliament and the Guide for the Care and Use of Laboratory Animals published by the US National Institutes of Health (NIH Publication No. 85–23, revised 1996). All mice were kept in humane conditions and animal studies were authorized by Ethics Committee of Shanghai University (Approval NO. ECSHU2020-0093) and reported in compliance with the ARRIVE guidelines.

### 4.12. Construction of the OVX Mouse Model and Drug Treatment

Under anesthesia with isoflurane gas, sham surgery or OVX was conducted in mice as previously described [37].

**Experiment** **1.**
*Thirty female 6-month-old C57BL/6J mice were divided into 5 groups. One week after surgery, acacetin (17.5 μmol/kg), EMCH-D6 (17.5 μmol/kg), or Acacetin-EMCH-D6 (17.5 μmol/kg) were intraperitoneally injected every 3 days over 5 weeks.*


**Experiment** **2.**
*Eighteen female 6-month-old C57BL/6J mice were divided into 3 groups. One week after surgery, Acacetin-EMCH-D6 (17.5 μmol/kg) and rapamycin (1 mg/kg) [38] were intraperitoneally injected every 3 d over 5 weeks.*


Animals were sacrificed at the end of the experiment by cervical dislocation.

Plasma was collected for the measurement of levels of C-terminal cross-linked telopeptides of type I collagen (CTX-I, NBP2-82444, Novus Biologicals, Littleton, CO, USA). The tibia and spines were harvested for micro-computed tomography (μCT) and Histological analysis.

### 4.13. Micro-Computed Tomography (μ. CT)

A three-dimensional reconstruction of the harvested bone samples including proximal tibia and L5 vertebra was created from images obtained using the Skyscan 1275 mini-micro CT scanner (BruKer, Karlsruhe, Germany) (voltage of 50 kV, current of 60 mA, and an isotropic resolution of 8 μm). The following parameters were calculated for trabecular bone and vertebrae: bone marrow density (BMD), the bone volume/tissue volume (BV/TV) ratio, trabecular separation (Tb. Sp), and the number of trabeculae (Tb.N) were analyzed.

### 4.14. Histological Analysis

For histopathological analysis, the proximal tibia was fixed in 4% paraformaldehyde. The paraffin-embedded tibial samples were cut into sections of 5-μm-thickness and stained with hematoxylin and eosin (H&E). TRAP staining was conducted for the assessment of differentiation and maturation of osteoclasts as described previously [39]. Bone sections were incubated with primary antibody against mouse Anti-LC3A/B (ab128025, Abcam) for measurement of autophagy.

### 4.15. Biophotonic Imaging Analysis

For the in vivo tracking process, FITC-labeled acacetin-EMCH (17.5 μmol/kg) and FITC-labeled Acacetin-EMCH-D6 (17.5 μmol/kg) were prepared in our lab and intraperitoneally injected into female 20-month-old C57BL/6J mice. Animals were sacrificed and relevant organs were harvested 0.5, 2, 6, and 12 h after injection. Finally, a Quickview 3000 system (Bio-Real Sciences, Salzburg, Austria) was utilized for fluorescence imaging of the distribution.

### 4.16. Statistical Analysis

All results were presented as mean ± SD. Comparison among groups was analyzed using the one-way ANOVA followed by Tukey’s post hoc analysis. Statistical analyses were conducted using GraphPad Prism version 9.0 software (GraphPad Software, San Diego, CA, USA). A *p*-value of <0.05 was used to denote statistical significance.

## 5. Conclusions

In conclusion, our study showed for the first time that Acacetin-EMCH-D6, as a novel acid-sensitive bone targeting delivery system carrying acacetin, was adequate for the treatment of postmenopausal osteoporosis in vivo. Mechanistically, the benefit of acacetin on osteoclast differentiation and bone resorption was partly dependent on its inhibitory effects on autophagy.

## Figures and Tables

**Figure 1 pharmaceuticals-16-00002-f001:**
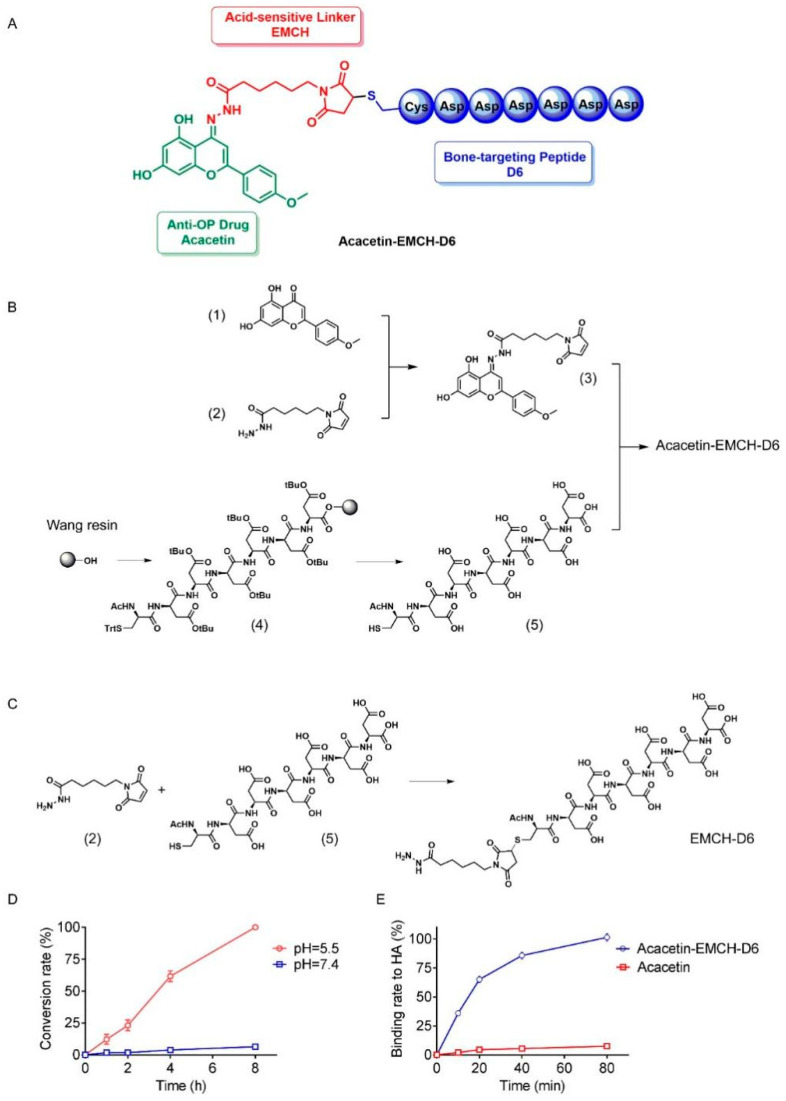
Synthesis and chemical characterization of Acacetin-EMCH-D6. Structure of Acacetin-EMCH-D6 (**A**). Chemical synthesis of the Acacetin-EMCH-D6 (**B**) and EMCH-D6 (**C**). The pH-dependent drug release assay of Acacetin-EMCH-D6 (**D**). Binding of Acacetin-EMCH-D6 and acacetin to hydroxyapatite (HA) (**E**). EMCH, N-ε-maleimidocaproic acid hydrazide.

**Figure 2 pharmaceuticals-16-00002-f002:**
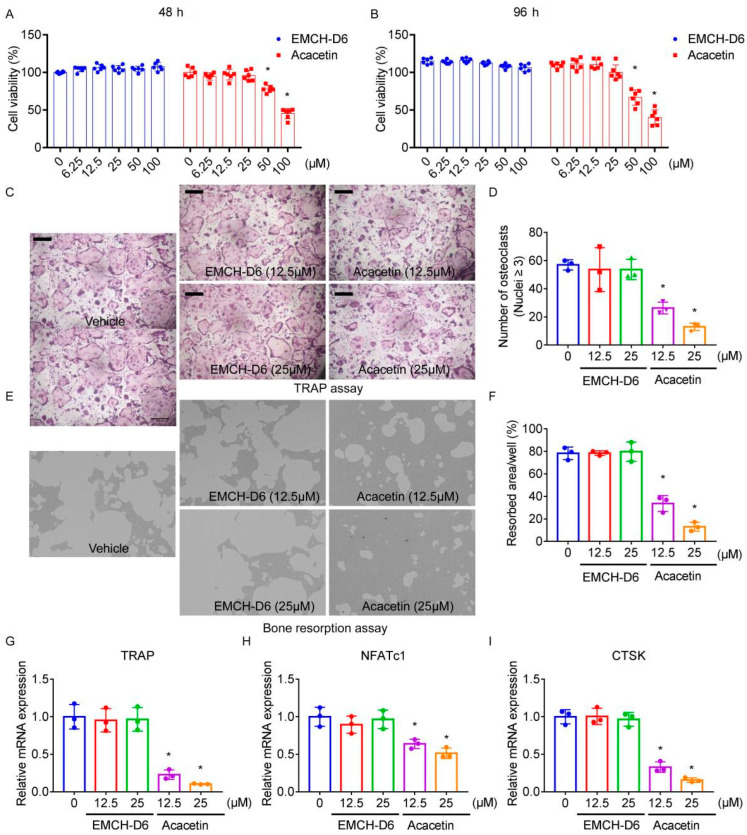
BMMs (8 × 10^3^ cells/well) were seeded to 96-well and treated with acacetin or EMCH-D6 (0, 6.25, 12.5, 25, 50, 100 μM) for 48 h (**A**) and 96 h (**B**), *n* = 6. The cytotoxic effects of acacetin or EMCH-D6 on BMMs proliferation were detected using the CCK-8 assay. BMMs were stimulated with RANKL (50 ng/mL) and M-CSF (25 ng/mL) in the presence of acacetin or EMCH-D6 (12.5, 25 μM) for 7 days. TRAP-positive multinucleated osteoclasts (≥3 nuclei) were counted using TRAP assay (**C**,**D**). BMMs (2 × 10^4^ cells/well) were seeded in Osteo Assay Surface plate and stimulated with RANKL (50 ng/mL) and M-CSF (25 ng/mL). After immature osteoclasts were formed, acacetin or EMCH-D6 (12.5, 25 μM) were added for 3 days. Bone resorption area was quantified (**E**,**F**). BMMs were stimulated with RANKL (50 ng/mL) and M-CSF (25 ng/mL) in the presence of acacetin or EMCH-D6 (12.5, 25 μM) for 3 days. The mRNA expression of TRAP (**G**), NFATc1 (**H**), and CTSK (**I**) were assayed by RT-PCR. * *p* < 0.05 versus the vehicle-treated group, *n* = 3.

**Figure 3 pharmaceuticals-16-00002-f003:**
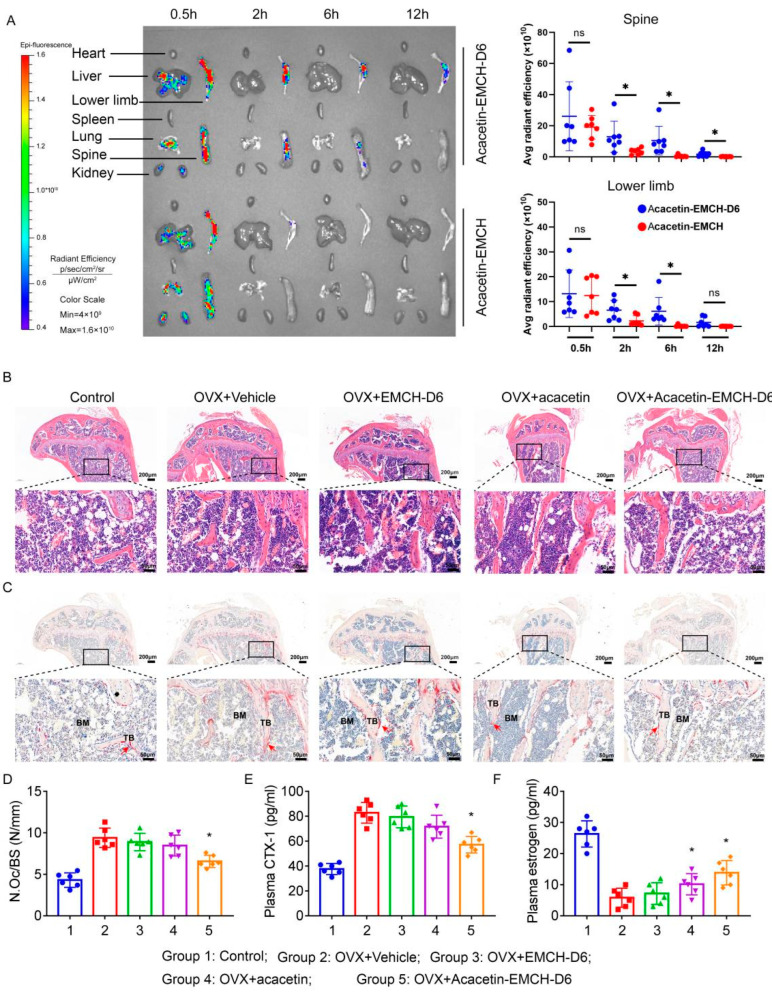
In vivo distribution of Acacetin-EMCH-D6 (**A**) Fluorescence of organs from mice sacrificed 30 min, 2 h, 6 h, and 12 h after intraperitoneal injection of FITC-labeled Acacetin-EMCH and Acacetin-EMCH-D6 (17.5 μmol/kg). One week after surgery of ovariectomy (OVX), mice were treated with Acacetin-EMCH-D6 or acacetin (i.p., 17.5 μmol/kg/3 days) for 5 weeks. H&E (**B**) and TRAP (**C**) staining were conducted on sections of proximal tibia. Representative microscopic images were shown and TRAP-positive osteoclast numbers (**D**) were calculated in each group. Graphs showed the levels of CTX-1 (**E**) and estrogen (**F**) in plasma. Sham-operated mice were used as control. * *p* < 0.05 versus the control group, *n* = 6.

**Figure 4 pharmaceuticals-16-00002-f004:**
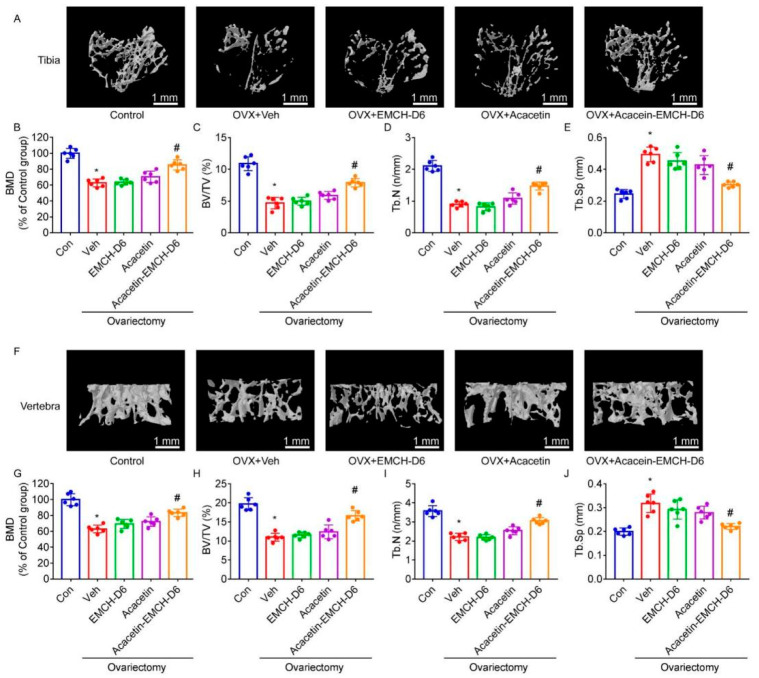
One week after surgery of ovariectomy (OVX), mice were treated with Acacetin-EMCH-D6 or acacetin (i.p., 17.5 μmol/kg/3 days) for 5 weeks. Left tibia and L5 vertebra were subjected to micro-CT scanning. Representative three-dimensional reconstructed images were presented (**A**,**F**). Trabecular structural parameters of proximal tibia and vertebrae including BMD (**B**,**G**), BV/TV (**C**,**H**), Tb.Th (**D**,**I**), Tb.N (**E**,**J**) were presented. * *p* < 0.05 versus the control group; # *p* < 0.05 versus the vehicle-treated OVX group, *n* = 6.

**Figure 5 pharmaceuticals-16-00002-f005:**
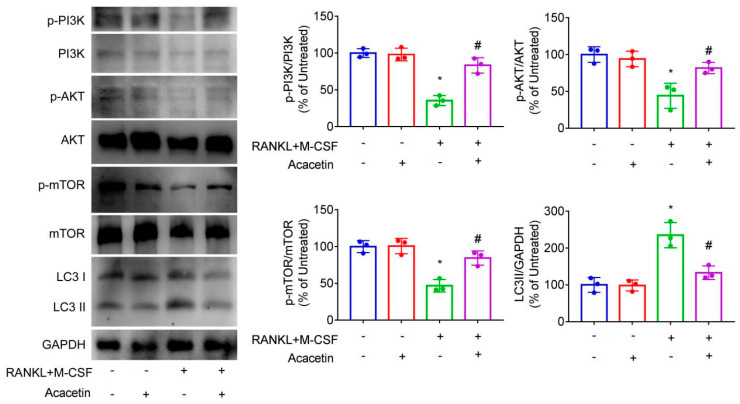
BMMs were stimulated with RANKL (50 ng/mL) and M-CSF (25 ng/mL) in the presence of acacetin (25 μM) for 6 h. Western blotting results and responding quantification of PI3K phosphorylation, AKT phosphorylation, mTOR phosphorylation, and LC3II expression were shown. * *p* < 0.05 versus the vehicle-treated group; # *p* < 0.05 versus the RANKL+M-CSF group, *n* = 3.

**Figure 6 pharmaceuticals-16-00002-f006:**
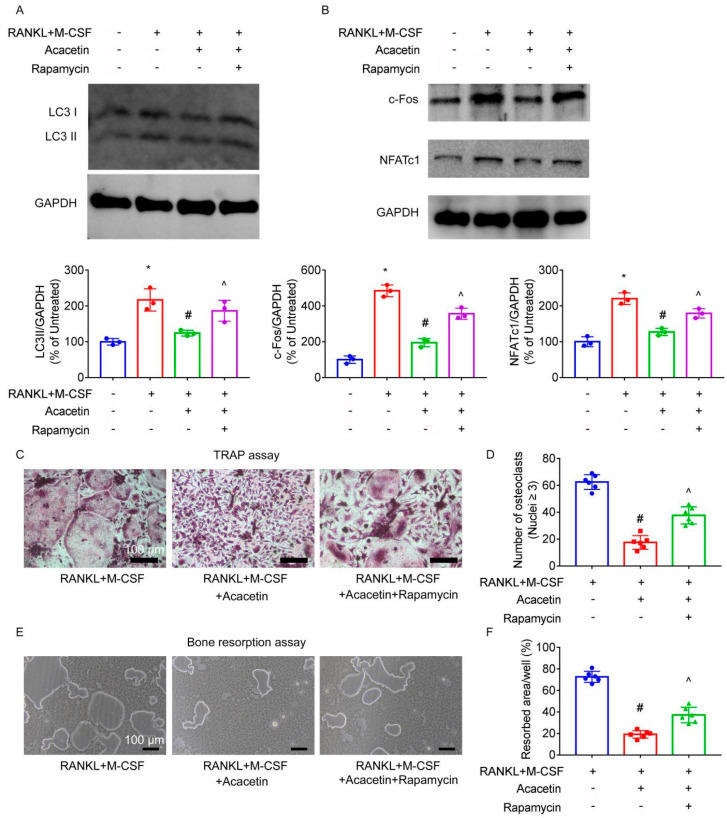
Activation of autophagy by rapamycin abolished the effect of acacetin on osteoclast differentiation and bone resorption in vitro. BMMs were stimulated with RANKL (50 ng/mL) and M-CSF (25 ng/mL) in the presence of acacetin (25 μM) and rapamycin (50 nmol/L) for 6 h. Protein expression of LC3 (**A**) was detected by western blotting analysis. BMMs were stimulated with RANKL (50 ng/mL) and M-CSF (25 ng/mL) in the presence of acacetin (25 μM) and rapamycin (50 nmol/L) for 3 days. Protein expression of c-Fos and NFATc1 (**B**) was detected by western blotting analysis. BMMs were stimulated with RANKL (50 ng/mL) and M-CSF (25 ng/mL) in the presence of acacetin (25 μM) and rapamycin (50 nmol/L) for 7 days. TRAP-positive multinucleated osteoclasts (≥3 nuclei) were counted using TRAP assay (**C**,**D**). BMMs were seeded in Osteo Assay Surface plate and stimulated with RANKL (50 ng/mL) and M-CSF (25 ng/mL). After immature osteoclasts were formed, acacetin (25 μM) and rapamycin (50 nmol/L) were added for 3 days. Bone resorption area was quantified (**E**,**F**). * *p* < 0.05 versus the vehicle-treated group; # *p* < 0.05 versus the RANKL+M-CSF group; ^ *p* < 0.05 versus the RANKL+M-CSF+acacetin group.

**Figure 7 pharmaceuticals-16-00002-f007:**
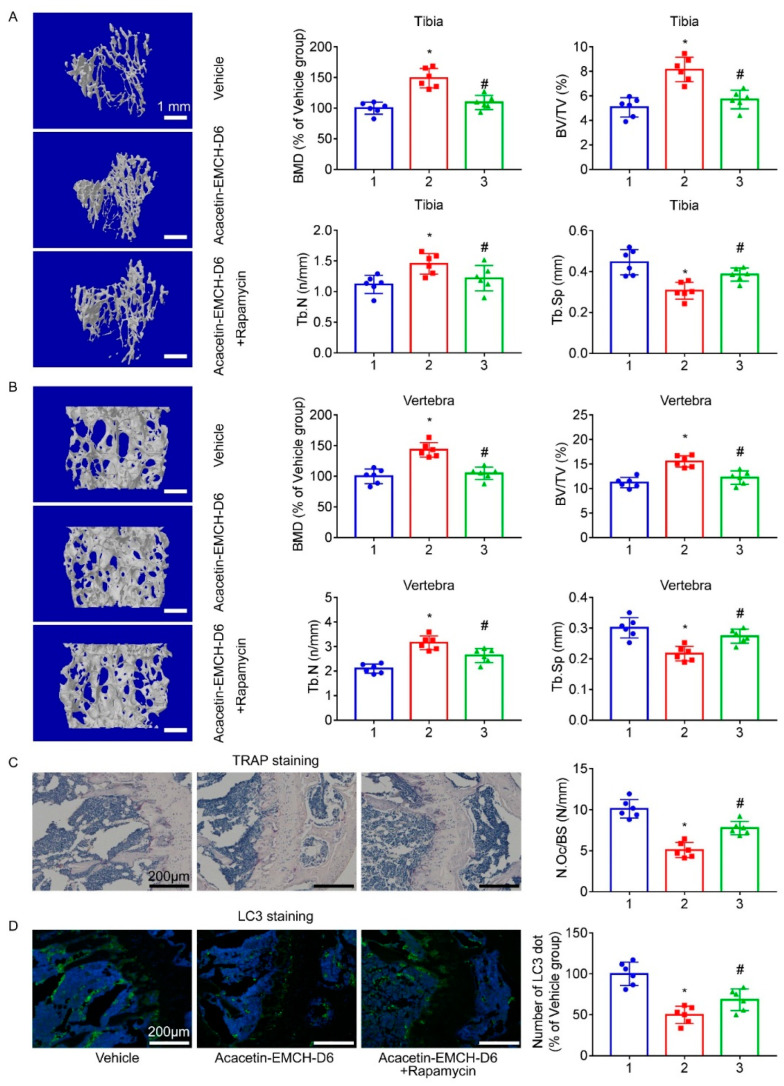
One week after surgery of ovariectomy (OVX), mice were treated with Acacetin-EMCH-D6 (17.5 μmol/kg/3 days, i.p.) and rapamycin (1 mg/kg/3 days, i.p.) for 5 weeks. Left tibia and L5 vertebrae were subjected to micro-CT scanning. Representative three-dimensional reconstructed images were presented (**A**,**B**). Trabecular structural parameters of proximal tibia and L5 vertebra including BMD, BV/TV, Tb.Th, and Tb.N were shown. TRAP (**C**) and LC3 (**D**) staining were conducted on sections of proximal tibia. Representative microscopic images were shown and numbers of TRAP-positive osteoclasts and LC3 dots were calculated in each group. * *p* < 0.05 versus the vehicle-treated OVX group; # *p* < 0.05 versus the Acacetin-EMCH-D6-treated OVX group.

**Table 1 pharmaceuticals-16-00002-t001:** Gene sequences of the mouse primer used in the study.

Genes	Sequence (5′–3′)	
*NFATc1*	Sense Antisense	CCGTTGCTTCCAGAAAATAACA TGTGGGATGTGAACTCGGAA
*CTSK*	Sense Antisense	CTTCCAATACGTGCAGCAGA TCTTCAGGGCTTTCTCGTTC
*TRAP*	Sense Antisense	TCCTGGCTCAAAAAGCAGTT ACATAGCCCACACCGTTCTC
*GAPDH*	Sense Antisense	ACCCAGAAGACTGTGGATGG CACATTGGGGGTAGGAACAC

## Data Availability

Data is contained within the article and Appendix A.

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
