# Peer review of "An Acid-Sensitive Bone Targeting Delivery System Carrying Acacetin Prevents Osteoporosis in Ovariectomized Mice"

_pharmaceuticals, 2022, doi:10.3390/ph16010002_

Round 1
Reviewer 1 Report
The paper reports on a bone targeting peptide drug conjugate that decreases osteoclast formation by inhibiting autophagy in OVX bone loss mice. This is an interesting approach. The data are numerous and nicely presented. However, the discussion part is extremely brief and superficial. I would suggest to elaborate much more on the outcomes and what can be learned from it. Also, explain the results much more in the context of the literature. Some literature is presented, but merely as one-sentence notes rather than how it relates to this study. Questions that should be answered in the discussion are: Why did you choose your mouse model? What do the proteins you investigated in your qPCR-study have to do with bone loss? How is the mTOR-pathway involved in autophagy and what role could be played by acatecin in this context? Other minor issues I put as comments into the pdf file.

Author Response
Response: We are sorry we did not express our meaning accurately.
Please explain shortly, why this peptide is specifically targeting bone.
Inspired by the fact that several bone noncollagenous proteins, such as osteopontin and bone sialoprotein, that bind to HA have a repeating sequence of acidic amino acids (Asp or Glu) in their structures as possible HA-binding sites. this peptide sequence was screened out by S Kasugai etc. in 2000 (J Bone Miner Res, 2010, 15, 936-943). The possible mechanism is that hydroxyl in hydroxyapatite can be replaced by carbonate ions in polyaspartic acid to form hydroxyapatite derivatives.
Suggestions reviewed in PDF were revised in manuscript and marked it in yellow.
Thank you a lot for giving us so much valuable suggestions and detailed directions. Each of your comments greatly helped us enhance our understanding towards this topic and improve the quality of this article. Thank you again, and hope that the correction will meet with approval.
Reviewer 2 Report
In the manuscript entitled “An acid-sensitive bone targeting delivery system carrying acacetin prevents osteoporosis in ovariectomized mice” by Sun et al. the authors describe the design of an acid-sensitive and bone-targeting system for the delivery of acacetin to the bone resorption site for the treatment of osteoporosis in the ovariectomy-induced osteoporosis murine model.
In addition, authors provide some mechanistically insights in the role of acacetin on inhibition of osteoclast autophagy.
Overall, the manuscript offers some interesting advances in the field of bone-targeted drug delivery, and, therefore, it should be considered within the scope of the journal Pharmaceuticals.
Several points, however, should be addressed before publication.
1. Line 55: “However, we did not observe its anti-osteoporotic effects at the dose of 5mg/kg/3 day in OVX mice”. Please, provide support for this statement
2. Line 86 “preparation of osteoblasts”. Provide characterization tests for osteoclasts isolation.
3. Line 139 and 142: “Experiment 1 and experiment 2”: define the number of animals per group
4. Line 149: “.μ. CT”. Better not to use abbreviation in subsection title.
5. Line 175: “P value of < 0.5 was used denote statistical significance.”. Revise.
6. Line 179: “According to literature reports, osteoclasts degrade HA at an acidic microenvironment to achieving bone remodeling processes.” Provide bibliographic information you are referring to.
7. Figure 2 A,B: what did you consider 100% in the cytotoxicity assay?
8. Line 487 and 88: check number of BMM cells per well seeded
9. Line 492 and 531: “Osteo Aaasy Surface”: revise
10. Line 237: “In lower limb and spine of Acacetin-EMCH-D6-injected group, fluorescence intensity was maintained at a high level 2 hrs after injection. 2 hrs after injection, little residual fluorescence was detected lower limb and spine of Acacetin-EMCH injected group. Our results indicated that EMCH-D6 facilitated acacetin targeting bone and prolonged the retention time in bone.” The statement refers to Fig 3A qualitatively showing 1 animal/group per timepoint: ideally authors should provide quantitative evaluation in a larger number of animals (on a selected timepoint).
Author Response
Comments to the Author
- Line 55: “However, we did not observe its anti-osteoporotic effects at the dose of 5mg/kg/3 day in OVX mice”. Please, provide support for this statement
- Line 86 “preparation of osteoblasts”. Provide characterization tests for osteoclasts isolation.
- Line 139 and 142: “Experiment 1 and experiment 2”: define the number of animals per group
- Line 149: “.μ. CT”. Better not to use abbreviation in subsection title.
- Line 175: “P value of < 0.5 was used denote statistical significance.”. Revise.
- Line 179: “According to literature reports, osteoclasts degrade HA at an acidic microenvironment to achieving bone remodeling processes.” Provide bibliographic information you are referring to.
- Figure 2 A,B: what did you consider 100% in the cytotoxicity assay?
- Line 487 and 88: check number of BMM cells per well seeded
- Line 492 and 531: “Osteo Aaasy Surface”: revise
- Line 237: “In lower limb and spine of Acacetin-EMCH-D6-injected group, fluorescence intensity was maintained at a high level 2 hrs after injection. 2 hrs after injection, little residual fluorescence was detected lower limb and spine of Acacetin-EMCH injected group. Our results indicated that EMCH-D6 facilitated acacetin targeting bone and prolonged the retention time in bone.” The statement refers to Fig 3A qualitatively showing 1 animal/group per timepoint: ideally authors should provide quantitative evaluation in a larger number of animals (on a selected timepoint).
Response: Thank you very much for pointing out this defect.
- We have provided data to support this statement in supplementary materials.
- In vitro maturation of macrophages into osteoclasts requires presence of macrophage colony-stimulating factor (M-CSF) and receptor for activation of nuclear factor kappa B (NF-κB) (RANK) ligand (RANKL) (also known as OPGL and TRANCE). MCSF, which is imperative for macrophage maturation, binds to its receptor, c-Fms, on early osteoclast precursors, thereby providing signals required for their survival and proliferation. Mature osteoclasts can be observed under a light microscope and stained with trap. (Teitelbaum SL. Bone resorption by osteoclasts.Science. 2000;289(5484):1504-1508. doi:10.1126/science.289.5484.1504)
- We have revised the number of animals per group in Experiment 1 and Experiment 2.
- We have revised “μCT”to “Micro-Computed Tomography”.
- We have revised “P value of < 0.05 was used denote statistical significance.”
- We have add reference in our manuscript.
- We use CCK-8 assay to detect OD value and converted into cell viability. 100*((experiment well mean value-blank well mean value)/(control well mean value-blank well mean value)).
- We have revised the number of BMM cell per well seeded.
- We have revised “Osteo Aaasy Surface”to “Osteo Assay Surface”.
- We have provided quantitative evaluation in a larger number of animals in revised figure3.
We really appreciate the time Reviewer2 has taken to review and revise our article. All of your comments were very helpful. We have carefully revised our article according to your instructions. Thank you again, and hope that our correction will meet with approval.
Reviewer 3 Report
This reviewer had no major technical and scientific issues with this manuscript and is in the opinion that it may be acceptable with a minor revision:
1. Please indicate which drug release model mechanism can best describe the release data presented in Figure 1.
Author Response
Comments to the Author
- Please indicate which drug release model mechanism can best describe the release data presented in Figure 1.
Response: Thank you very much for pointing out this defect.
Figure1 indicated that pH-dependent drug release study.
Drug was dissolved in 4 mM sodium phosphate buffer containing 150 mM sodium chloride (pH=7.4 or 5.5) to obtain a final concentration of 800 μM. The samples were incubated at room temperature and measured by HPLC at t=0, 1, 2, 4, and 8 h.
We tried our best to improve the manuscript and made some changes in the manuscript. These changes will not influence the content and framework of the paper but make the article more logical and accurate.
Reviewer 4 Report
In the presented article Authors have described the novel acid-sensitive bone targeting delivery system comprising acacetin for the treatment of postmenopausal osteoporosis in vivo. In the light of current, global problems related to osteoporosis, I consider this study valuable and adding some value to the current state of knowledge. My critical remarks are related only to the editorial and linguistic aspects. Please check the entire text carefully for typos, e.g. abstract, line 34 - 'bone reportion sites', different font sizes, e.g. page 4, lines 178-185, page 5, lines 186-200 and other language errors. With these slight corrections, I can recommend the following manuscript for publication in Pharmaceuticals.
Author Response
Comments to the Author
In the presented article Authors have described the novel acid-sensitive bone targeting delivery system comprising acacetin for the treatment of postmenopausal osteoporosis in vivo. In the light of current, global problems related to osteoporosis, I consider this study valuable and adding some value to the current state of knowledge. My critical remarks are related only to the editorial and linguistic aspects. Please check the entire text carefully for typos, e.g. abstract, line 34 - 'bone reportion sites', different font sizes, e.g. page 4, lines 178-185, page 5, lines 186-200 and other language errors. With these slight corrections, I can recommend the following manuscript for publication in Pharmaceuticals.
Response: Thank you very much for pointing out this defect.
- We have revised “bone reportion sites”to “bone resorption sites”.
- We have revised font sizes in page 4 lines 178-185, page 5, lines 186-200 and other language errors.
We sincerely appreciate for all of your warm work, and hope that the correction will meet with approval.
Once again, thank you very much for your comments and suggestions.